# Colloid driven low supersaturation crystallization for atomically thin Bismuth halide perovskite

Lutao Li[1,3], Junjie Yao[1,3], Juntong Zhu[1,3], Yuan Chen[2,3], Chen Wang[2], Zhicheng Zhou[1], Guoxiang Zhao[1], Sihan Zhang[1], Ruonan Wang[1], Jiating Li[1], Xiangyi Wang[1], Zheng Lu[1], Lingbo Xiao[1], Qiang Zhang[2] & Guifu Zou ®[1] ✉

It is challenging to grow atomically thin non-van der Waals perovskite due to the strong electronic coupling between adjacent layers. Here, we present a colloid-driven low supersaturation crystallization strategy to grow atomically thin $Cs_3Bi_2Br_9$. The colloid solution drives low-concentration solute in a supersaturation state, contributing to initial heterogeneous nucleation. Simultaneously, the colloids provide a stable precursor source in the low-concentration solute. The surfactant is absorbed in specific crystal nucleation facet resulting in the anisotropic growth of planar dominance. Ionic perovskite $Cs_3Bi_2Br_9$ is readily grown from monolayered to six-layered $Cs_3Bi_2Br_9$ corresponding to thicknesses of 0.7, 1.6, 2.7, 3.6, 4.6 and 5.7 nm. The atomically thin $Cs_3Bi_2Br_9$ presents layer-dependent nonlinear optical performance and stacking-induced second harmonic generation. This work provides a concept for growing atomically thin halide perovskite with non-van der Waal structures and demonstrates potential application for atomically thin single crystals' growth with strong electronic coupling between adjacent layers.

Atomically thin single crystals have promising applications for optoelectronics, ferroelectrics, ferromagnetics, piezoelectrics, and superconducting fields[1–8]. When the thickness of materials is reduced to atomically thin sizes, the broken electron wave function dimension and interlayer coupling interaction significant impact on the intrinsic properties, involving the change of band-gap and superconducting transition temperature, from metal to semiconductor transition, from indirect to direct semiconductor transition, and from anti-ferromagnetism towards ferromagnetism transition[8–12]. Atomically thin single crystals are of fundamental significance to the study of physical properties. The development of two-dimensional (2D) atomically thin semiconductors is not only conducive for exploring the relationships between structure and properties but also for deriving insights for requisite synthetic strategy. Lead-free halide perovskites have drawn much attention recently due to their

environmental friendliness, excellent stability, and remarkable photoelectric properties[13–16]. They are considered as an ideal substitutes for lead halide perovskites in future optoelectronic devices[17–20]. Recently, scientists have demonstrated their superiority in white light illumination[16,21], high-definition displays[22–24], solar cells[25–29], photodetectors[15,30], transistors[31], photocatalysis[32]. $Cs_3Bi_2Br_9$, in particular, is a typical lead-free halide perovskite[24,33,34], that is derived from cubic $CsPbBr_3$ with a trivalent bismuth substituted bivalent lead by rebalancing the total charge and the low energy structure via lattice reconstruction. Although $Cs_3Bi_2Br_9$ has been widely explored as quantum dots, polycrystalline films, and bulky single crystals in the past[15,35–39], the atomically thin $Cs_3Bi_2Br_9$ with high aspect ratios have not been reported due to the strong inter-layered interaction.

Typical methods, such as chemical vapor deposition and mechanical exfoliation, have been successful in growing various kinds

[1]College of Energy, Key Laboratory of Advanced Carbon Materials and Wearable Energy Technologies of Jiangsu Province, Soochow University, Suzhou 215006, China. [2]College of Mechanical and Electronic Engineering, Shandong University of Science and Technology, Qingdao 266590, China. [3]These authors contributed equally: Lutao Li, Junjie Yao, Juntong Zhu, Yuan Chen. ✉e-mail: zouguifu@suda.edu.cn

of atomically thin single crystals with inter-layered van der Waals interaction[40–42]. Nevertheless, the connections among octahedral layers of $Cs_3Bi_2Br_9$ are ionic bonds with strong Coulomb force instead of weak van der Waals interaction[33,43–45], which is different from the traditional 2D van der Waals crystals (such as graphene, transition metal dichalcogenides, and Ruddlesden–Popper phase perovskites)[1,46,47]. It is necessary to find a strategy for achieving atomically thin single crystal without inter-layered van der Waals interaction. One promising method is colloid synthesis to grow nanosheets or quantum dots with the thicknesses down to even a single cell[48–52]. It is suitable for the growth of layered and non-layered materials[49,51,52]. However, the colloid synthesis usually suffers from undesired explosive nucleation, smaller lateral size, and difficulty separating single crystals, which limits further research on atomically thin single crystal[19,49]. In recent years, researchers have made many efforts to synthesize lead-free atomically thin halide perovskite[34,48,53–56]. Nevertheless, there is no report on the growth of high aspect ratio atomically thin halide perovskite with non-van der Waal structures through a chemical solution strategy.

Here, we introduce a colloid driven low supersaturation crystallization strategy for the growth of atomically thin $Cs_3Bi_2Br_9$. The colloid formation establishes a low supersaturation solution, which produces a low nucleation density. Simultaneously, the colloids provide a stable solute source for kinetic growth. The surfactant is absorbed in the (0001) crystal facet, resulting in an anisotropic growth of planar dominance. The thickness of atomically thin $Cs_3Bi_2Br_9$ is feasibly regulated from monolayered to six-layered $Cs_3Bi_2Br_9$. The atomically thin $Cs_3Bi_2Br_9$ presents layer-dependent nonlinear optical performance and stacking-induced second harmonic generation (SHG). This work provides a concept for growing atomically thin non-van der Waal inter-layered halide perovskite crystals.

## Results

According to the crystal structure, the in-plane layer of $Cs_3Bi_2Br_9$ is constructed by $[BiBr_6]^{3-}$ octahedra and the connection of corner sheared Br atoms. The out-of-plane layers are connected by Cs-Br ionic bonds with a larger bond length of 4.05 Å (Supplementary Fig. 1a), resulting in a relatively weak force between adjacent layers[33]. However, the bond energy of Cs-Br is calculated to be 2.815 eV (or 270.69 kJ/mol), which is stronger than the van der Waals interaction (< 0.104 eV or 10 kJ/mol)[57]. Therefore, bulk crystals are easily produced rather than thin platelets (Supplementary Fig. 1b). To overcome the contradiction, a colloid solution is proposed to drive low nucleation density and secure the anisotropic growth through a low supersaturation crystallization strategy. The clear diagram of growth process can be described in Fig. 1a–e. Initially, the original precursor solution is uniformly distributed (Fig. 1a). The solution concentration for $Cs^+$ and $Bi^{3+}$ at room-temperature can be changed from about $10^2$ mM to 1 mM as the n-OA percentage adds from 0% to 95% (Supplementary Fig. 2). The n-OA is added in the original solution to form a colloidal solution with Tyndall effect. According to Fajans rule, the most of precursor ion is absorbed in the colloids and less ions remain in the bulk solution (Fig. 1b). The structure of the colloidal solution can be described by the Stern double layer (typical colloids' structure) in Supplementary Fig. 3. The low supersaturation colloidal solution is aimed to enhance the heterogeneous nucleation on the substrate. Supersaturation plays crucial role in sustaining continuous crystal growth and prevent atomically thin samples from dissolving. As the temperature increases, the adsorption strength of the ions of colloidal solution weakens, allowing ions absorbed by the colloids to have higher kinetic energy and move towards the surrounding heterogeneous nuclei on the substrate. That is, colloid solution provides the precursor source in the low-concentration solute system to grow on the substrate (Fig. 1c). Moreover, the surfactant is directionally adsorbed on the $Cs_3Bi_2Br_9$ specific crystal facet (0001) to regulate the

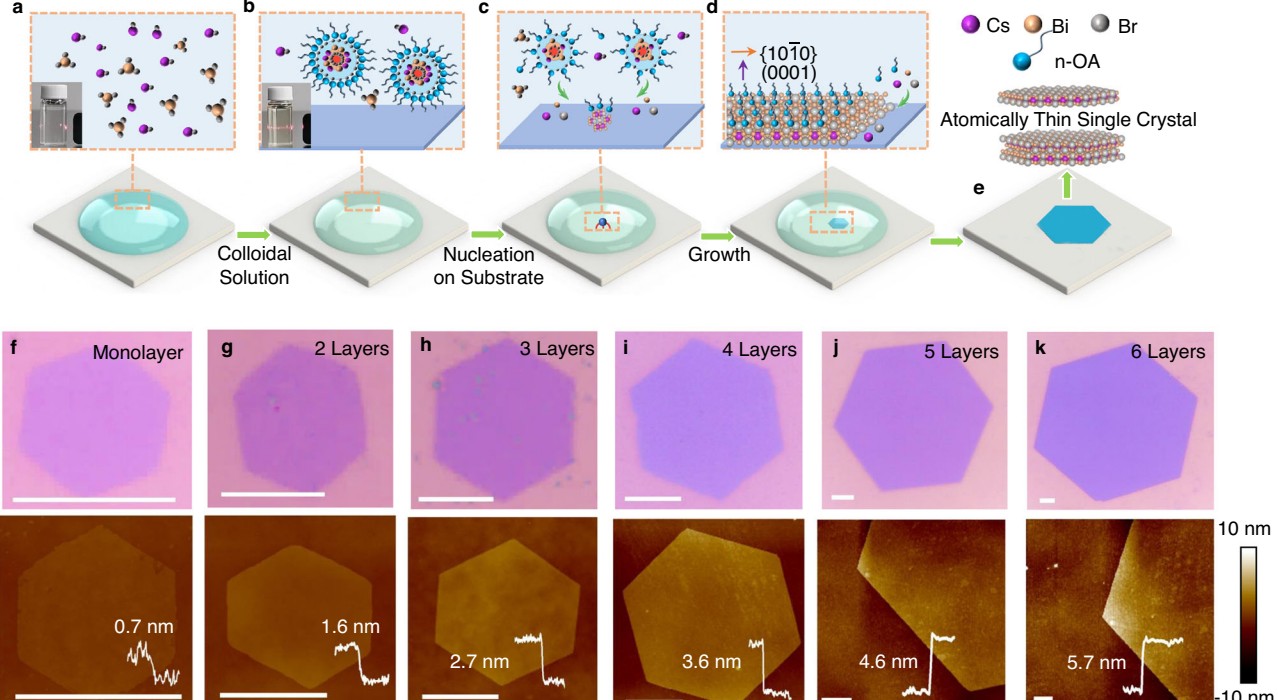

**Fig. 1 | The colloid driven low supersaturation for atomically thin $Cs_3Bi_2Br_9$.**
**a** Initially, the original precursor solution is uniformly distributed. **b** The most of precursor ion is absorbed in the colloids and less ions remain in the bulk solution. **c** Colloid solution provides the precursor source in the low-concentration solute system to grow on the substrate. **d** The surfactant is directionally adsorbed on the $Cs_3Bi_2Br_9$ specific crystal facets to promote anisotropic growth. **e** The atomically thin $Cs_3Bi_2Br_9$ are obtained on the substrate. **f–k** Optical microscopy images and corresponding AFM images of the obtained few layer $Cs_3Bi_2Br_9$, all of the scale bars are 5 μm. The AFM images share the same thickness scales on the right side. The inset is the corresponding thick pattern, and all of the scale bars are 5 μm. Source data are provided as a Source Data file.

surface energy and promote anisotropic growth (Fig. 1d). In the case of the organic acid surfactant, the binding of carboxyl group (-COOH) ligands typically involves hydrogen bonding between the protons located on the protic ligand's anchoring group and the Br anions, with carbonyl group (C=O) pointing toward the positive Bi sites, respectively (Supplementary Figs. 4, 5)[58]. Finally, atomically thin $Cs_3Bi_2Br_9$ are grown on the substrate (Fig. 1e).

The optical microscopy images of the as-produced atomically thin $Cs_3Bi_2Br_9$, as shown in Fig. 1f–k, depict regular hexagons. The color of the atomically thin $Cs_3Bi_2Br_9$ shows significant contrast, indicating an increased thickness. Corresponding thicknesses can be estimated to be about 0.7, 1.6, 2.7, 3.6, 4.6 and 5.7 nm from atomic force microscopy (AFM) measurements (Fig. 1f–k). The simulated height of the $Cs_3Bi_2Br_9$ single cell is around 0.67 nm, and the layer spacing is around 0.33 nm (P-3m1, cif#2106376). The above thickness values could correspond to the single to six cells. It is worthy of noting that the small surface roughness of 0.16 nm suggests a clean and smooth platelet surface (Supplementary Fig. 6). Compared to the previous reports of atomically thin halide perovskite (Supplementary Fig. 7), the $Cs_3Bi_2Br_9$ possesses the largest aspect ratio (size/thickness) over $10^4$, which is beneficial for the application of atomically thin halide perovskite materials in optoelectronic devices and electronic device integration. The size and thickness of $Cs_3Bi_2Br_9$ are influenced by the substrates, which may be due to lattice matching. The experimental results show the $Cs_3Bi_2Br_9$ on sapphire substrate of good lattice match tends to smaller thickness and larger aspect ratios (Supplementary Fig. 8). In general, the heterogeneous nuclei can bind more strongly to the substrate, resulting in the formation of atomically thin single crystal on the substrate surface[59]. Additionally, it is found that the wettability of the substrate also affects the growth of atomically thin $Cs_3Bi_2Br_9$. The atomically thin $Cs_3Bi_2Br_9$ grown on ozone-treated substrates exhibits larger lateral sizes and aspect ratios compared to those grown on untreated substrates (Supplementary Fig. 9). Obviously, the ozone-treated substrates have smaller contact angles than these of untreated substrates. The hydrophilic substrate enhances heterogeneous nucleation to produce bigger lateral sizes of atomically thin $Cs_3Bi_2Br_9$.

The temperature plays a crucial role in controlling the growth kinetics process. As the temperatures is increased from 60 °C to 100 °C, the thickness and lateral size of the platelets exhibit a noticeable decrease (Supplementary Fig. 10a–d). The corresponding thicknesses of the platelets are measured at approximately 12 - 20 cells, 6 - 15 cells, 4 - 12 cells, 1 - 10 cells, respectively. When the temperature reaches 100 °C, the few and/or single cell platelets could be easily obtained, and the corresponding lateral size is several to ten micrometers (Supplementary Fig. 10e–h). It is clearly demonstrated that the lateral sizes of the platelet continuously decrease with the increasing temperature, while the thickness first decreases and then increases with temperature, and the turning point is around 100 °C (Supplementary Fig. 10i, j). Furthermore, the nucleation density presents an increasing trend with the temperature (Supplementary Fig. 10k), indicating a close dependence on thermodynamics. Following the regulation of the temperature, we further adjust the amount of surfactant in the experiment process (Supplementary Fig. 11). Atomically thin $Cs_3Bi_2Br_9$ can be realized under the surfactant concentration ($R_{n-OA}$ = n-OA/(n-OA + DMSO)) from 35% to 50% (Fig. 2a–h). Based on surfactant experiments, Fig. 2i shows the statistics of the lateral size and thickness. As the amount of surfactant increases, the thickness of the atomically thin $Cs_3Bi_2Br_9$ decreases significantly, and the lateral size of $Cs_3Bi_2Br_9$ slightly decreases. It is helpful to achieve a large aspect ratio of atomically thin $Cs_3Bi_2Br_9$. At a concentration of 40%, the thickness of $Cs_3Bi_2Br_9$ can be achieved below 5 nm, and the lateral size can reach over 50 μm, which presents a high aspect ratio over $10^4$. The large lateral size, combined with the atomically thin thickness, is beneficial for further application for optoelectronic devices and device integration. While the amount of surfactant over 45%, it is hard to

obtain complete hexagonal atomically thin $Cs_3Bi_2Br_9$ due to the low precursor concentration.

To further understand the surfactant mechanism, density functional theory is utilized to calculate the surface energy. The surface energies of (0001) and {10$\bar{1}$0} bare planes in DMSO solvent ($\gamma_{DMSO}$) are estimated to be −46.35 eV and −15.46 eV, respectively (Fig. 2j, k left). This suggests an anisotropic structure in the in-plane and out-of-plane directions. The surface energies of (0001) and {10$\bar{1}$0} in the n-OA solvent ($\gamma_{n-OA}$) are calculated to be −178.25 eV and −146.8 eV, respectively (Fig. 2j, k right). This illustrates that n-OA molecular passivation significantly reduces the surface energy and increases the activation energy barrier (Fig. 2j, k middle). Simultaneously, the corresponding adsorption energies ($E_{adsorp}$) of (0001) and {10$\bar{1}$0} are around −1.14 eV and −0.58 eV, respectively (Fig. 2j, k middle), indicating that the (0001) facet more easily adsorbs the n-OA molecule and results in edge growth dominant along the <10$\bar{1}$0> directions. Consequently, the surfactant is directionally adsorbed on the $Cs_3Bi_2Br_9$ specific crystal facet (0001) to regulate the surface energy and promote anisotropic growth along the <10$\bar{1}$0> directions. In addition to n-OA, various organic molecules (including acetic acid, n-butyric acid, n-valeric acid, n-hexanoic acid, undecanoic acid, and oleic acid) with the same carboxyl functional group were tested as replacements for n-OA in the growth of atomically thin $Cs_3Bi_2Br_9$. Supplementary Fig. 12 displays that the short chain molecules are more conducive to the crystal growth, and the long chain molecules are unfavorable for $Cs_3Bi_2Br_9$ crystallization. It may derive from the viscosity, boiling point, and solubility.

The structure and composition of the atomically thin $Cs_3Bi_2Br_9$ are investigated in detail. The X-ray diffraction pattern presents in Supplementary Fig. 13, which periodic peaks are located at two thetas of 9.04°, 18.08°, 27.24°, 46.10°, 56.06°, respectively. According to JCPDS No. 44-0714 (space group: P-3m1, $a = b = 8.216$ Å, $c = 10.0698$ Å; $\alpha = \beta = 90°$, $\gamma = 120°$), which corresponds to the (0001), (0002), (0003), (0005), and (0006) facets, respectively. A transmission electron microscope was used to investigate the inner microstructure of the atomically thin $Cs_3Bi_2Br_9$. The atomically thin $Cs_3Bi_2Br_9$ was transferred onto microgrid copper mesh after growth on PMMA, and PMMA was removed by chlorobenzene immersion. The precise atomic structure is investigated by high-angle annular dark field scanning transmission electron microscopy (HAADF-STEM). Figure 3a displays precise and long-range ordered atomic arrangement without visible point, line, and plane defects, which demonstrates that the atomically thin $Cs_3Bi_2Br_9$ is a single crystal with high quality. The lattice spacing is estimated to be 0.41 nm in two directions, which corresponds to the (10$\bar{1}$2) and (10$\bar{1}$2) lattice facets. According to the enlarged HAADF and IDPC images (Fig. 3b, c), the atomic image can be seen with different contrasts (white, grey, and black). The nuclear radius can be distinguished with the size of white > grey > black. From the difference in atomic radius, they could be easily attributed to Cs, Bi, and Br atoms, respectively (Fig. 3b insert). The simulated atomic structure of the $Cs_3Bi_2Br_9$ P-3m1 phase is shown in Fig. 3d, which is in good agreement with the experimental results (Fig. 3b, c). In addition, the selected area electron diffraction pattern presents a clear lattice with typical P-3m1 space group features, which further demonstrates that the sample is well-defined single crystal (Fig. 3e). The low magnification image of the platelet in Fig. 3f presents highly transparent and clear outlines under electron beam irradiation. The corresponding energy dispersive X-ray spectrometer mapping images show Cs, Bi, and Br elements distributed uniformly. Furthermore, the precise atomic arrangement along the side view is also investigated and shown in Fig. 3g, h, which presents that the clear Cs+Bi, Bi, and Br atomic columns arrange at the side view, and the distance of adjacent Cs+Bi atomic column is measured to be 1.01 nm. All the data agree well with the simulated atomic structure.

To understand the crystal structure of atomically thin $Cs_3Bi_2Br_9$, the thickness-dependent Raman spectra of the atomically thin $Cs_3Bi_2Br_9$

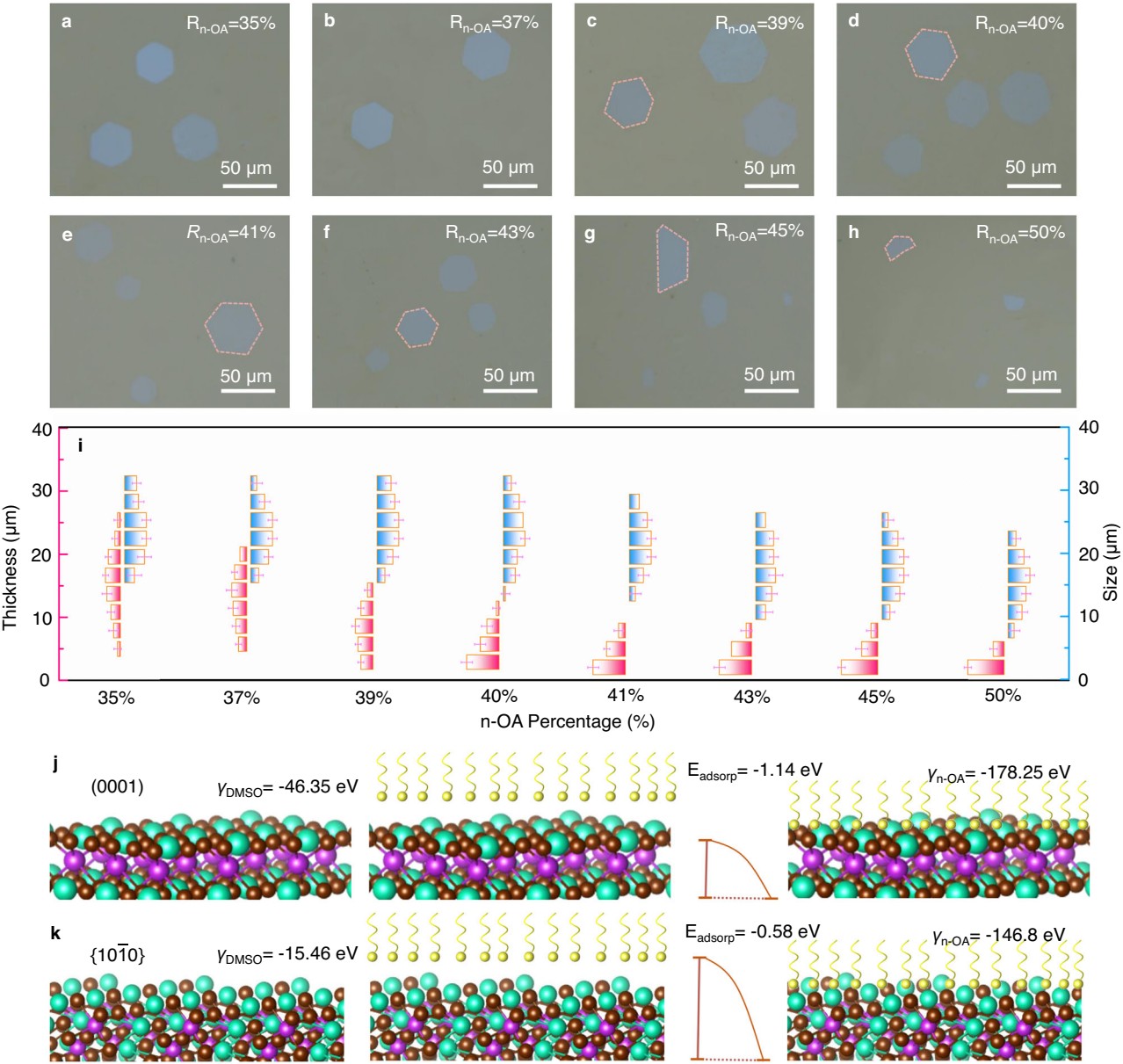

**Fig. 2 | The growth regulation of atomically thin Cs₃Bi₂Br₉.** Optical microscopy images of the atomically thin Cs₃Bi₂Br₉ that produced from ranged n-OA percentage: (**a**) 35%, (**b**) 37%, (**c**) 39%, (**d**) 40%, (**e**) 41%, (**f**) 43%, (**g**) 45%, and (**h**) 50%. **i** The statistics of Cs₃Bi₂Br₉ thicknesses and lateral sizes as a function of ranged n-OA percentage. The thickness and lateral size error bars represent the standard deviation. **j** Illustration and DFT calculation of the surface energy on the Cs₃Bi₂Br₉ bare (0001) facet, n-OA adsorption energy on the (0001) facet, and surface energy of the (0001) facet with n-OA passion. **k** Illustration and DFT calculation of the surface energy on the Cs₃Bi₂Br₉ bare {10$\bar{1}$0} facet, n-OA adsorption energy on the (1010) facet, and surface energy of the {10$\bar{1}$0} facet with n-OA passion. Source data are provided as a Source Data file.

are shown in Fig. 4a. There are two strong and sharp peaks located at 165.8 cm$^{-1}$ and 191.4 cm$^{-1}$ with a uniform distribution throughout the entire platelet (Supplementary Fig. 14). They are attributed to $E_g$ and $A_{1g}$ resonance, respectively. $E_g$ is the in-plane vibrational mode, and $A_{1g}$ is the out-of-plane vibrational mode of the [BiBr₆]³⁻ octahedron (as illustrated in the insert of Fig. 4b, c)[60]. The stretching modes of the BrBi₂ groups give rise to a very weak depolarized Raman band at 89.8 cm$^{-1}$ $A_g$ (v$_s$BiBr₂). The bands below 80 cm$^{-1}$ are related to the deformation modes of the Br–Bi arrangements and the translational modes of Cs$^+$ [60]. $A_{1g}$ gradually moves towards a higher wavenumber (blueshift) as the platelet thickness decreases from 30 nm to about 2.7 nm (Fig. 4b). $E_g$ gradually blueshifts as the thickness decreases from 50 nm to 2.7 nm (Fig. 4c). This suggests a stronger interlayer interaction and the lattices constantly shrink as the thickness of the Cs₃Bi₂Br₉ decreases. The variations of $E_g$ and $A_{1g}$ are statistics as a function of

thickness (Fig. 4b, c). To summarize, the Raman peaks are decreased from 163.0 cm$^{-1}$ to 167.8 cm$^{-1}$ and from 185.7 cm$^{-1}$ to 192.4 cm$^{-1}$ as the platelet thickness decreases from 50 nm to two cells thickness, which could be used as a simple means to determine the thickness of the platelets.

Due to the non-centrosymmetric structure of Cs₃Bi₂Br₉, Fig. 5a shows a nonlinear optical application, where a 1064 nm wavelength laser is used as the excitation source for SHG. Supplementary Fig. 15 displays the SHG intensity regularly increases as the laser intensity increasing from 25.7 mW to 152.5 mW. The SHG intensity is performed as a function of excitation power, which is well-fitted in logarithmic function coordinates with a slope of 1.79, which is close to the theoretical value of 2 from the electric dipole approximation theory[61]. Atomically thin layers may exhibit mutations of photophysical properties, particularly as the number of layers reduces to a single layer[16,62].

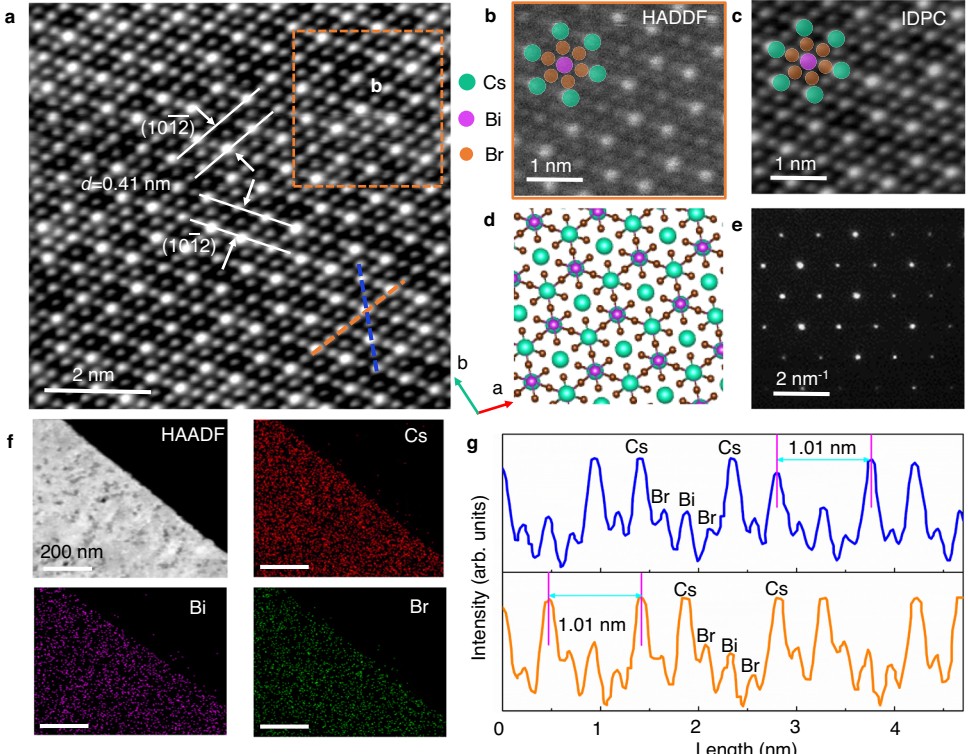

**Fig. 3 | Composition and structure characterization of atomically thin Cs₃Bi₂Br₉.** **a** HAADF-STEM image of the Cs₃Bi₂Br₉. **b** Enlarged HAADF-STEM image of the region in **a**. **c** IDPC image of the region in **a**. **d** Simulated atomic structure of Cs₃Bi₂Br₉ view along the (001) direction. **e** Selected area electron diffraction image of the Cs₃Bi₂Br₉. **f** STEM energy dispersive spectrometer mapping images, the elements are Cs, Bi, and Br, respectively. The scale bar is 200 nm. **g** Intensity line profile along the blue and orange dashed lines in **a**. Source data are provided as a Source Data file.

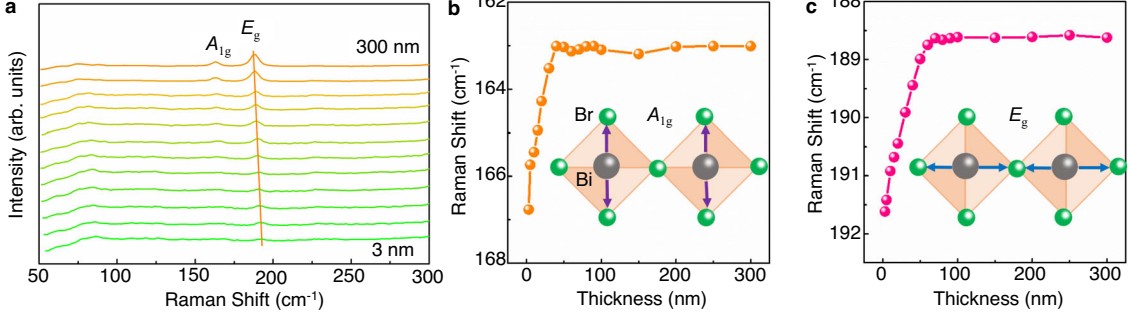

**Fig. 4 | The thickness-dependent Raman spectra of atomically thin Cs₃Bi₂Br₉.** **a** Raman spectra of the atomically thin Cs₃Bi₂Br₉ with a thickness range from 300 nm to 2.67 nm. **b** Statistical analysis of the $A_{1g}$ vibration as a function of the thickness range from 300 nm to 2.67 nm. The insert illustrates the $A_{1g}$ vibration in the [BiBr₆]³⁻ octahedron. **c** Statistics of the $E_g$ vibration as a function of the thickness range from 300 nm to 2.67 nm. The insert illustrates the $E_g$ vibration in the [BiBr₆]³⁻ octahedron. Source data are provided as a Source Data file.

Due to the three-fold rotational symmetry in the few-layer Cs₃Bi₂Br₉ crystal structure, the co-polarized SHG intensity displays a six-fold rotational symmetry as a function of Cs₃Bi₂Br₉'s azimuthal angle θ (Fig. 5b). We systematically measured SHG intensity for Cs₃Bi₂Br₉ crystals with 1–10 layers. SHG intensity exhibits different symmetry properties between the odd- and even-layer. Strong SHG signal can be detected for all odd-layers, while even-layers show weak SHG signal (Fig. 5c). This is because that even-layers of Cs₃Bi₂Br₉ belong to the centrosymmetric D₃d space group, whereas odd-layer number belongs to the non-centrosymmetric D₃h space group. Furthermore, the SHG signal is reduced with the increase of the odd-layer number, which may be attributed to interlayer coupling. The variation of SHG intensity with odd- and even-layers shows the potential application in nonlinear

optics for atomically thin Cs₃Bi₂Br₉. As it is well known, artificial structures created by stacked two-dimensional crystals have been the focus of intense research activity[63]. As for twisted or stacked graphene layers[64] and transition metal dichalcogenides layers[65], these structures can show unusual behaviors and phenomena. Among the various layered compounds exhibit interesting properties governed by their structural symmetry and interlayer coupling, which are highly susceptible to stacking[66]. Figure 5d, e shows very weak SHG intensity of the region 1 (6-layered Cs₃Bi₂Br₉), while the region 3 (7-layered Cs₃Bi₂Br₉) has higher SHG intensity. This may be caused by the SHG intensity contrast to the different symmetry properties between Cs₃Bi₂Br₉ samples with odd- and even-layer. The corresponding AFM images of the region in Fig. 5d is shown in Supplementary Fig. 16. SHG

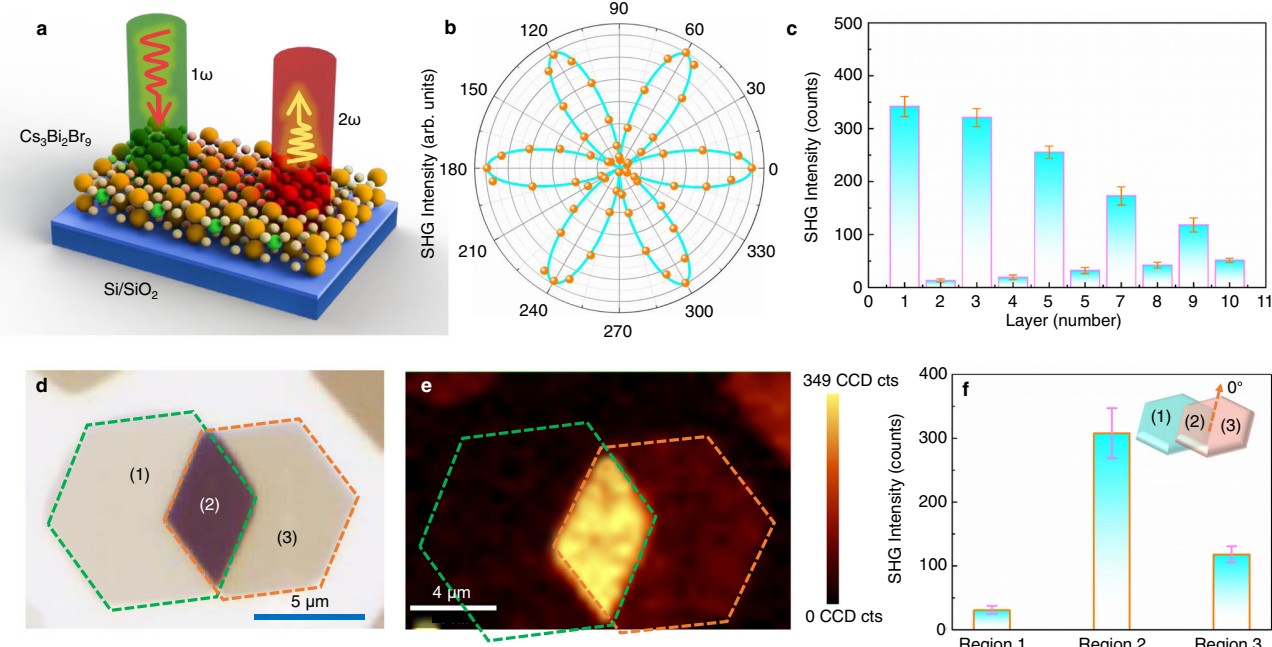

**Fig. 5 | SHG characterization of atomically thin $Cs_3Bi_2Br_9$. a** Schematic diagram of SHG measurement of atomically thin $Cs_3Bi_2Br_9$. **b** Polarization-resolved SHG spectra of the a three-layered $Cs_3Bi_2Br_9$, revealing its underlying threefold symmetry. **c** Layer-dependent SHG verified the systematically stronger signals from odd-layer $Cs_3Bi_2Br_9$ due to inversion symmetry breaking than from even-layer $Cs_3Bi_2Br_9$. The SHG intensity error bars represent the standard deviation. **d** The optical images of the stacking 6-layered and the 7-layered $Cs_3Bi_2Br_9$ crystal. **e** The SHG intensity mapping of the region in **a**. **f** The SHG intensity of the three regions. The inset diagram shows three regions with stacking angle of $\theta = 0°$. The SHG intensity error bars represent the standard deviation. Source data are provided as a Source Data file.

emissions show very bright signal in region 2, which may be due to the 3R-like stacking with the vertical stacking angle $\theta = 0°$ (Fig. 5f). The total SHG intensity in stacking region can be expressed as[67]

$$I_S(\theta) = I_1 + I_2 + 2\sqrt{I_1 I_2}\cos 3\theta \tag{1}$$

where $I_S$, $I_1$, and $I_2$ stand for the SHG intensity in the stacking region, the 6-layered $Cs_3Bi_2Br_9$ and the 7-layered $Cs_3Bi_2Br_9$, respectively. $\theta$ is the stacking angle ($\theta = 0°$). The SHG intensity in the stacking region is well fitted with this Eq. (1). These SHG findings on atomically thin halide perovskites would be beneficial for fundamental research and potential application in the field of electronics and optoelectronics.

## Discussion

A colloid-driven low supersaturation crystallization strategy is proposed to realize the growth of atomically thin $Cs_3Bi_2Br_9$. Colloid formation leads to a low supersaturation solution, which is responsible for heterogeneous nucleation on the substrate and kinetics dominant anisotropic growth. The colloidal solution in the subsequent growth process provides a stable precursor source, ensuring continuous growth. With the directional absorption of surfactant and edge-dominated growth, the thickness of thin platelets could be adjusted from monolayer to few layers. The resulting atomically thin $Cs_3Bi_2Br_9$ presents layer-dependent optical performance and stacking-induced SHG. This work provides a concept for growing atomically thin single crystal platelets of non-van der Waal inter-layer perovskites.

## Methods
### Growth of atomically thin $Cs_3Bi_2Br_9$

The atomically thin $Cs_3Bi_2Br_9$ is synthesized from a modified drop casting and solvent evaporation crystallization process. Briefly, 15 mM CsBr and 10 mM $BiBr_3$ are firstly dissolved in 1 mL dimethyl sulfoxide to form a transparent solution. Then, 0.5 mL n-Octanoic acid is added, and the transparent solution becomes turbid, the solution is centrifuged at speed 4000 g for 5 min to remove the insoluble matter. Subsequently, 20 μL of the above solution is dropped onto a cleaned $SiO_2$/Si substrate and heated to 100 °C for 10 min, accompanied by nucleation and growth. Finally, the substrate is immersed in toluene solvent to remove the n-octanoic acid and unreacted metal halide salts.

### Characterization

Optical microscopy images are taken using a Nikon ECLIPSE LV150N microscope; atomic force microscopy is taken from Bruker Dimension Icon; scanning electron microscopy images are taken from a Hitachi (SU-8010); transmission electron microscopy and scanning transmission electron microscopy images are recorded from (F20, 200 KV) which equipped with X-ray energy dispersion spectra; X-ray diffraction patterns are taken in an X-ray diffractometer (Bruker D8 Advance). The Raman spectrum is measured by a Raman spectrometer (Horiba Jobin Yvon HR Evolution, the excitation light source is a 532 nm monochromatic laser). The X-ray photoelectron spectroscopy is recorded from Thermo Fisher Escalab 250Xi. The FITR spectrum is measured from Bruker Nicolet IS50.

### Computational method

The generalized gradient approximation of Perdew-Burke-Ernzerhof within the framework of DFT was carried out on the basis of all-electron-like projector-augmented wave potentials, as implemented in the Vienna Ab-initio Simulation Package. A plane-wave cut-off energy of 400 eV is used for the self-consistent calculations.

### Statistics and reproducibility

Statistical analysis of thicknesses and lateral sizes was all performed by five or more samples on different substates with same fabrication conditions. Statistical analysis of SHG intensity was performed by five or more $Cs_3Bi_2Br_9$ platelets on the same substrate. Results are shown as mean ± standard deviation. No data were excluded from the statistical analyses.

**Reporting summary**

Further information on research design is available in the Nature Portfolio Reporting Summary linked to this article.

## Data availability

The data that support the findings of this study are available within the article and its Supplementary Information files or from the corresponding author upon request. Source data are provided in this paper. Source data are provided with this paper.

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

## Acknowledgements
We gratefully acknowledge the National Natural Science Foundation of China (21971172), the Priority Academic Program Development (PAPD) of Jiangsu Higher Education Institutions for Optical Engineering, the Key Lab of Advanced Optical Manufacturing Technologies of Jiangsu Province & Key Lab of Modern Optical Technologies of Education Ministry of China in Soochow University, and the Jiangsu Collaborative Innovation Center of Photovoltaic Science and Engineering in Changzhou University.

## Author contributions
L.L., J.Y., J.Z., and Y.C. contributed equally. The manuscript was written through the contributions of all authors. G.Zou supervised the whole project. L.L., Y.C., and J.Y. conceived the idea, designed the experiments, and analyzed the data. J.Z., C.W., Z.Z., G.Zhao, S.Z., R.W., J.L., X.W., Z.L., and L.X. assisted in the material characterizations. Q.Zhang participated in discussions and provided helpful suggestions. L.L., Y.C., and G.Zou wrote the paper.

## Competing interests
The authors declare no competing interests.
