## [Peer Review File · Nature Communications]

REVIEWERS' COMMENTS:

Reviewer #1 (Remarks to the Author):

Comments on the manuscript entitled " Colloid Driven Low Supersaturation Crystallization for Atomically Thin Bismuth Halide Perovskite Single Crystal Films" (NCOMMS-23-04190) by Lutao Li et al.

This manuscript outlines a novel approach for growing perovskite single-crystal nanosheets using a low supersaturation crystallization strategy that relies on colloidal solutions and surfactants. With this technique, the authors can synthesize inorganic perovskite Cs₃Bi₂Br₉ with varying thicknesses, independent of the substrate used. The paper provides a detailed synthesis process, material characterization, and a comprehensive explanation of the growth mechanism. Additionally, the resulting single-crystal nanocrystals exhibit excellent nonlinear optical properties, expanding their potential applications in the field of optoelectronics. The manuscript is well-written and includes insightful discussions, making it suitable for publication in Nature Communications with only minor revisions.

The detailed comments are as followings:

1. Certain nanomaterials, including antimonene and MoS₂ (Adv. Mater. 2018, 30, 1803244; Angew. Chem. 2015, 127, 3155), exhibit distinct photophysical properties at different atomic layers, particularly as the number of layers reduces to a single layer, which may lead to mutations. It would be valuable for the author to explore and discuss the changes in photophysical properties that arise as a function of layer number.
2. The growth of single crystal nanosheets results in variations in size and thickness when grown on different substrates. What factors might contribute to these differences?
3. The author tested various additives. Have any of them been found to induce growth in alternate directions? Specifically, is it possible to alter the growth direction of perovskite single crystals by employing surfactants that match other crystal planes?

Reviewer #2 (Remarks to the Author):

Atomically thin single crystal films are of fundamental significance to the physical properties study. For lead-free perovskite, it is challenging to grow the atomically thin single crystal films. In the present work, the authors claim that they developed a strategy to grow atomically thin Cs₃Bi₂Br₉ single crystal film. The key of this method is the colloid formation and surfactant adsorption on specific facet. Additionally, they studied the component-dependent polarity second harmonic generation of the samples. However, attention should be paid to the following points:

1. Actually, the resulting samples are not single crystal films, but micro-platelets. So the synthesis is colloidal synthesis of micro- or nano-particles.
2. The property of polarity second harmonic generation is also not new for metal halide materials.
3. The authors claim that the colloidal particles are thermodynamically unstable and can release metal salt ions for the growth of micro-platelets. What is the content of the colloidal particles. It is also possible that the particles aggregate to form the micro-platelets.
4. How does the substrate affect the synthesis?
5. How to know the surfactant (carboxyl group) is binding with the Br atom based on the XPS and IR spectrum (Figure S2 and S3), please give detailed explanation.

Therefore, I cannot suggest publishing the present manuscript on Nature Communications.

Reviewer #3 (Remarks to the Author):

The work "Colloid Driven Low Supersaturation Crystallization for Atomically Thin Bismuth Halide Perovskite Single Crystal Films" reports on the synthesis and characterization of Cs₃Bi₂Br₉ thin

platelets. The authors demonstrated that, by tuning the amount of n-octanoic acid (n-OA) in the reaction mixture, they could tune the lateral size and the thickness of the obtained platelets. Higher amount of n-OA allows to obtain single-cell-thick crystals.

The authors demonstrated also that the material displays non-linear optical properties.

The manuscript has several flaws, in my opinion; it is definitely not acceptable for publication on Nature Communications.

First, it is difficult to read - it might require a deep revision of the used English.

Second, it is hard to identify the specific novelty of the work (maybe because of the first issue on language): reports on the colloidal synthesis of Bi-based "perovskites" and on their non linear optical properties can be found already in literature. The authors should highlight better how their approach differs from those of other research groups.

Apart from these two main issues, there also other problems:

1) the authors used the word "film" to describe their platelets.

2) the majority of the paper is focused on discussing concepts as supersaturation in colloidal synthesis, that might most likely be skipped.

3) the authors claim that carboxyl functional groups bind to Br atoms, based on XPS. It's hard to imagine how two negatively-charged ions are kept together.

4) is the thickness scale for the AFM images of Figure 1 the same for all panels?

5) The histograms in Figure 2 are not discussed in the text nor in the figure caption. Lateral size and thickness statistics suggest moreover that the used method produces objects non really homogeneous in size.

RESPONSE TO REVIEWERS' COMMENTS

Dear Reviewers,

We are very grateful to the reviewers for his/her time and effort to review our manuscript. We are extremely encouraged by the feedback from all of the reviewers. We thank the reviewers for their constructive comments, which have helped us to improve our manuscript further.

To address the comments/suggestions, we have carried out more experiments and measurements to improve the manuscript. The changes we have made are summarized below. These changes are blue-marked in the revised manuscript as well. More results are compiled in "supporting information" submitted with the manuscript. The whole revisions are based on all the comments from the reviewers, so we believe that the data from the additional experiments can fully address the reviewers' concerns.

Reviewer#1:

This manuscript outlines a novel approach for growing perovskite single-crystal nanosheets using a low supersaturation crystallization strategy that relies on colloidal solutions and surfactants. With this technique, the authors can synthesize inorganic perovskite $\text{Cs}_3\text{Bi}_2\text{Br}_9$ with varying thicknesses, independent of the substrate used. The paper provides a detailed synthesis process, material characterization, and a comprehensive explanation of the growth mechanism. Additionally, the resulting single-crystal nanocrystals exhibit excellent nonlinear optical properties, expanding their potential applications in the field of optoelectronics. The manuscript is well-written and includes insightful discussions, making it suitable for publication in Nature Communications with only minor revisions.

A: Thank reviewer#1 for your positive comments on our work. According to your valuable suggestions, we have revised the manuscript as followings.

The detailed comments are as followings:

1. Certain nanomaterials, including antimonene and MoS_2 (*Adv. Mater.* **2018**, 30, 1803244; *Angew. Chem.* **2015**, 127, 3155), exhibit distinct photophysical properties at different atomic layers, particularly as the number of layers reduces to a single layer, which may lead to mutations. It would be valuable for the author to explore and discuss the changes in photophysical properties that arise as a function of layer number.

A: According to the reviewer's suggestion, we have added second harmonic generation (SHG) measurements to explore mutations of photophysical properties in revised manuscript (Page 9, Line 3-29; Page 10, Line 1-3). Atomically thin layers may exhibit mutations of photophysical properties, particularly as the number of layers reduces to a single layer. SHG measurements were carried out by the atomically thin $\text{Cs}_3\text{Bi}_2\text{Br}_9$ to explore the photophysical properties. Due to the three-fold rotational symmetry in the few-layer $\text{Cs}_3\text{Bi}_2\text{Br}_9$ crystal structure, the co-polarized SHG intensity displays a six-fold rotational symmetry as a function of $\text{Cs}_3\text{Bi}_2\text{Br}_9$'s azimuthal angle θ (Figure R1a). We systematically measured SHG intensity for $\text{Cs}_3\text{Bi}_2\text{Br}_9$ crystals with 1-10 layers. SHG intensity exhibits different symmetry properties between the odd- and even-layer. Strong SHG signal can be detected for all odd-layers, while even-layers show weak SHG signal (Figure R1b). It could be concluded that even-layers of $\text{Cs}_3\text{Bi}_2\text{Br}_9$ belong to the centrosymmetric D_{3d} space group, whereas odd-layer number belongs

to the non-centrosymmetric D_{3h} space group. In addition, the SHG signal is reduced with the increase of the odd-layer number, which may be attributed to interlayer coupling. The variation of SHG intensity with odd- and even-layers shows the potential application in nonlinear optics for atomically thin $Cs_3Bi_2Br_9$.

Figure R1 (a) Polarization-resolved SHG spectra of the a three-layered $Cs_3Bi_2Br_9$, revealing its underlying threefold symmetry. (b) Layer-dependent SHG verified the systematically stronger signals from odd-layer $Cs_3Bi_2Br_9$ due to inversion symmetry breaking than from even-layer $Cs_3Bi_2Br_9$.

As it is well known, artificial structures created by stacked two-dimensional crystals have been the focus of intense research activity (*Nature* **2013**, 499, 419). As for twisted or stacked graphene layers (*Nature* **2013**, 497, 598) and transition metal dichalcogenides layers (*Science* **2022**, 376, 973), these structures can show unusual behaviors and new physical properties. Among the various layered compounds exhibit interesting properties governed by their structural symmetry and interlayer coupling, which are highly susceptible to stacking (*Nat. Nanotechnol.* **2014**, 9, 825). Herein, we found new SHG phenomena and interesting results in the field of perovskite's materials. Figure R2a-d shows very weak SHG intensity of the region 1 (6-layered $Cs_3Bi_2Br_9$), while the region 3 (7-layered $Cs_3Bi_2Br_9$) has higher SHG intensity. This may be caused by the SHG intensity contrast to the different symmetry properties between $Cs_3Bi_2Br_9$ samples with odd- and even-layer. SHG emissions show very bright signal in region 2, which may be due to the 3R-like stacking with the vertical stacking angle $\theta=0^\circ$ (Figure R2e). The total SHG intensity in stacking region can be expressed as:

$$I_S(\theta) = I_1 + I_2 + 2\sqrt{I_1 I_2} \cos 3\theta$$

where I_S , I_1 , and I_2 stand for the SHG intensity in the stacking region, the 6-layered $\text{Cs}_3\text{Bi}_2\text{Br}_9$ and the 7-layered $\text{Cs}_3\text{Bi}_2\text{Br}_9$, respectively. θ is the stacking angle ($\theta=0^\circ$, Figure R2f). The SHG intensity in the stacking region is well fitted with this equation (Figure R2e). This new phenomenon is the first observation in the field of metal halide materials. These SHG findings on atomically thin halide perovskites would be beneficial for fundamental research and potential application in the field of electronics and optoelectronics.

Figure R2 (a) The optical images of the stacking 6-layered and the 7-layered $\text{Cs}_3\text{Bi}_2\text{Br}_9$. (b) The SHG intensity mapping of the region in a. (c) The AFM images of the region in a. (d) The height of the marked line in region a. (e) The SHG intensity of the three regions. (f) The diagram with three regions with stacking angle of $\theta=0^\circ$.

2. The growth of single crystal nanosheets results in variations in size and thickness when grown on different substrates. What factors might contribute to these differences?

A: According to the reviewer's suggestion, we have included more experiments to investigate the growth of atomically thin $\text{Cs}_3\text{Bi}_2\text{Br}_9$ nanosheets on different substrates in revised manuscript (Page 5, Line 23-29; Page 6, Line 1-3). The different substrates have an influence on the size and thickness (Figure R3), which might be caused by lattice match. As it is well known, the lattice spacing values of in Al_2O_3 along the $\langle 10\bar{1}0 \rangle$ direction ($\frac{\sqrt{3}}{2} a_{\text{Al}_2\text{O}_3} = 0.412 \text{ nm}$), which can match well with the lattice spacing values in $\text{Cs}_3\text{Bi}_2\text{Br}_9$ ($a_{\text{Cs}_3\text{Bi}_2\text{Br}_9} = 0.822 \text{ nm}$) by building the supercells. The experimental results

show the $\text{Cs}_3\text{Bi}_2\text{Br}_9$ on sapphire substrate of well lattice match tends to smaller thickness and larger aspect ratios (Figure R3c-c'). In general, the heterogeneous nuclei can bind more strongly to the substrate. Heterogeneous nucleation is promoted on substrate, while homogeneous nucleation is suppressed, resulting in atomically thin halide perovskite being formed on the substrate surface layer-by-layer (*Chem. Soc. Rev.* **2019**, 48, 2011). In addition, we find that the wettability of the substrate also affects the growth of atomically thin $\text{Cs}_3\text{Bi}_2\text{Br}_9$ nanosheets. The atomically thin $\text{Cs}_3\text{Bi}_2\text{Br}_9$ grown on ozone-treated substrates exhibits larger lateral sizes and aspect ratios compared to those grown on untreated substrates (Figure R3a'-d').

Figure R3 The atomically thin $\text{Cs}_3\text{Bi}_2\text{Br}_9$ growing on untreated (a) mica, (b) SiO_2 , (c) Si and (d) sapphire substrates. The atomically thin $\text{Cs}_3\text{Bi}_2\text{Br}_9$ growing on ozone-treated (a') mica, (b') SiO_2 , (c') Si and (d') sapphire substrates.

Figure R4a-d presents the wettability of the untreated mica, SiO_2 , Si and sapphire substrates. Obviously, the ozone-treated substrates have smaller contact angles than these of untreated substrates (Figure R4a'-d'). Figure R4e illustrates an interfacial energy diagram for three phases of two solids and a liquid in contact. The terms γ_{cl} , γ_{sl} and γ_{cs} represent interfacial energies between the crystal phase and the liquid, the solid surface and the liquid, and the crystal phase and the solid surface, respectively. The nuclei and active centers have a high affinity with the solid surface. And the nucleation barrier (ΔG_{hetero}) is dramatically lowered due to effective reduction in the interface energy. The free energy for heterogeneous nucleation is corrected by introducing Φ , which is the ratio of free energies of homogeneous and heterogeneous nucleation (Eq. R1). Φ is a factor dependent on the contact angle θ

(Eq. R2).

$$\Delta G_{hetero} = \Phi \Delta G_{hetero} \quad (R1)$$

$$\Phi = \frac{(2 + \cos\theta)(1 - \cos\theta)^2}{4} \quad (R2)$$

The relationship between Φ and θ indicates that heterogeneous nucleation readily occurs at a low concentration state (Figure R4f). In order to grow atomically thin $\text{Cs}_3\text{Bi}_2\text{Br}_9$ with large aspect ratio, it is important to form heterogeneous nuclei on the substrate rather than homogeneous nucleation (*Chem. Soc. Rev.* **2019**, 48, 2011). As a result, the hydrophilic substrate enhances heterogeneous nucleation to produce bigger lateral sizes of atomically thin $\text{Cs}_3\text{Bi}_2\text{Br}_9$.

Figure R4 The wettability of the untreated (a) mica, (b) SiO_2 , (c) Si and (d) sapphire substrates. The wettability of the ozone-treated (a') mica, (b') SiO_2 , (c') Si and (d') sapphire substrates. (e) Illustration of the contact angle θ for heterogeneous nucleation. (f) Ratio of free energies of homogeneous and heterogeneous nucleation Φ as a function of the contact angle θ .

3. The author tested various additives. Have any of them been found to induce growth in alternate directions? Specifically, is it possible to alter the growth direction of perovskite single crystals by employing surfactants that match other crystal planes?

A: Yes, it could alter the growth direction of lead-free perovskite single crystals by employing

surfactants that match other crystal planes. The previous reference also reports employing organic amines that match (100) crystal planes can alter the growth direction of Cs₃Bi₂I₉ single crystals (*ACS Energy Lett.* **2022**, 7, 3370). Herein, several organic acids with varying chain lengths are used in Cs₃Bi₂Br₉ growth, including acetic acid, butyric acid, valeric acid, caproic acid, octanoic acid, undecanoic acid, and oleic acid. Due to the low boiling point of short-chain organic acids resulting in rapid nucleation, it tends to form the small grains instead of atomically thin single crystals. On the other hand, long-chain acids have higher steric hindrance to obstruct the growth process of atomically thin single crystals, and long-chain acids have a high boiling point resulting in difficulty of being removed after growth. Therefore, octanoic acid of moderate chain is chosen as the surfactant in Cs₃Bi₂Br₉ growth process. As for the organic acids, the binding of carboxyl group -COOH ligands usually involves hydrogen bonding between the protons located on the protic ligand's anchoring group and the Br anions, with carbonyl group C=O pointing toward the positive Bi sites, respectively (*ACS Nano* **2022**, 16, 1444), which might result in binding on specific facets.

Reviewer#2:

Atomically thin single crystal films are of fundamental significance to the physical properties study. For lead-free perovskite, it is challenging to grow the atomically thin single crystal films. In the present work, the authors claim that they developed a strategy to grow atomically thin $\text{Cs}_3\text{Bi}_2\text{Br}_9$ single crystal film. The key of this method is the colloid formation and surfactant adsorption on specific facet. Additionally, they studied the component-dependent polarity second harmonic generation of the samples. However, attention should be paid to the following points.

A: Thank you for your constructive comments. We have added more experiments to improve the manuscript according to the reviewer#2's request.

1. Actually, the resulting samples are not single crystal films, but micro-platelets. So the synthesis is colloidal synthesis of micro- or nano-particles.

A: According to the reviewer's comment, we have removed the "single crystal films" in the revised manuscript. Herein, the colloidal solution (Figure R5) is neither colloid synthesis nor colloid solid particles, but is used to build up a low supersaturation solution, which means low concentration of ions in bulk solution and more ions in the colloid. Low concentration of ions aims to be no nucleation in the bulk solution but less nucleation on the substrate, and supersaturation could continually provide ions' source from colloids for the atomically thin $\text{Cs}_3\text{Bi}_2\text{Br}_9$ growth. During the experiment process, we observe an obvious Tyndall effect (Figure R5a). Based on the typical colloids' structure described by the Stern double layer model (Electrochemistry Principles, Methods and Applications, *Oxford Univ. Press*, Oxford, 1993; *J. Colloid Interf. Sci.* 2007, 305, 159), Figure R5b shows the colloidal diagram of the Stern double layer. The nucleus particle could be negatively charged, attracting positive ions to form the Stern layer, where ions are strongly bound to nucleus particle. The diffusion layer contains a large amount of solvent and ions, and the ions are loosely attached the Stern layer. The diffusion layer can continuously exchange ions with the bulk solution. According to the zeta potential, the colloidal solution system is stable (Figure R5c). As a result, the very low concentration of ions remains in bulk solution, many ions are included in the colloidal structure. As-prepared low supersaturation solution is aimed to reduce the nucleation in bulk solution, and enhance the heterogeneous nucleation on the substrate. While temperature increases, the ions source of colloidal solution continually release for the growth of the atomically thin $\text{Cs}_3\text{Bi}_2\text{Br}_9$ on the substrate with the help of surfactants. The atomically

thin $\text{Cs}_3\text{Bi}_2\text{Br}_9$ growth is substrate-dependent heterogeneous nucleation process. The colloid solution provides a stable precursor source and create a low-concentration solute system. It drives a low-concentration solute to initialize heterogeneous nucleation on the substrate not in the bulk solution. The similar heterogeneous nucleation and growth process has been described by the Frank-van der Merwe model by the layer-by-layer growth (*Chem. Soc. Rev.* **2019**, 48, 2011).

Figure R5 (a) The Tyndall effect of original solution and colloidal solution by adding n-OA. (b) The Stern double layer model of the colloidal solution. (c) The zeta potential of $\text{Cs}_3\text{Bi}_2\text{Br}_9$ colloidal solution in H_2O at 25°C .

2. The property of polarity second harmonic generation is also not new for metal halide materials.

A: Except for the normal second harmonic generation of metal halide materials, we found some new SHG phenomena and interesting results on nonlinear optical properties of atomically thin $\text{Cs}_3\text{Bi}_2\text{Br}_9$. Atomically thin $\text{Cs}_3\text{Bi}_2\text{Br}_9$ shows layer-dependent SHG intensity (Figure R6) and stacking-induced SHG intensity (Figure R7). The vertical stacking phenomenon is the first observation in the field of metal halide materials. The specific experiment and analysis are addressed as followings.

We have added new second harmonic generation (SHG) measurements to explore mutations of photophysical properties in revised manuscript (Page 9, Line 3-29; Page 10, Line 1-3). Atomically thin layers may exhibit mutations of photophysical properties, particularly as the number of layers reduces to a single layer. SHG measurements were carried out by the atomically thin $\text{Cs}_3\text{Bi}_2\text{Br}_9$ to explore the photophysical properties. Due to the three-fold rotational symmetry in the few-layer $\text{Cs}_3\text{Bi}_2\text{Br}_9$ crystal structure, the co-polarized SHG intensity displays a six-fold rotational symmetry as a function of $\text{Cs}_3\text{Bi}_2\text{Br}_9$'s azimuthal angle θ (Figure R6a). We systematically measured SHG intensity for $\text{Cs}_3\text{Bi}_2\text{Br}_9$ crystals with 1-10 layers. SHG intensity exhibits different symmetry properties between the odd- and even-layer. Strong SHG signal can be detected for all odd-layers, while even-layers show weak SHG

signal (Figure R6b). It is because that even-layers of $\text{Cs}_3\text{Bi}_2\text{Br}_9$ belong to the centrosymmetric D_{3d} space group, whereas odd-layer number belongs to the non-centrosymmetric D_{3h} space group. In addition, the SHG signal is reduced with the increase of the odd-layer number, which may be attributed to interlayer coupling. The variation of SHG intensity with odd- and even-layers shows the potential application in nonlinear optics for atomically thin $\text{Cs}_3\text{Bi}_2\text{Br}_9$.

Figure R6 (a) Polarization-resolved SHG spectra of the a three-layered $\text{Cs}_3\text{Bi}_2\text{Br}_9$, revealing its underlying threefold symmetry. (b) Layer-dependent SHG verified the systematically stronger signals from odd-layer $\text{Cs}_3\text{Bi}_2\text{Br}_9$ due to inversion symmetry breaking than from even-layer $\text{Cs}_3\text{Bi}_2\text{Br}_9$.

As it is well known, artificial structures created by stacked two-dimensional crystals have been the focus of intense research activity (*Nature* **2013**, 499, 419). As for twisted or stacked graphene layers (*Nature* **2013**, 497, 598) and transition metal dichalcogenides layers (*Science* **2022**, 376, 973), these structures can show unusual behaviors and new physical properties. Among the various layered compounds exhibit interesting properties governed by their structural symmetry and interlayer coupling, which are highly susceptible to stacking (*Nat. Nanotechnol.* **2014**, 9, 825). Herein, we found new and interesting results. Figure R7a-d shows very weak SHG intensity of the region 1 (6-layered $\text{Cs}_3\text{Bi}_2\text{Br}_9$), while the region 3 (7-layered $\text{Cs}_3\text{Bi}_2\text{Br}_9$) has higher SHG intensity. This may be caused by the SHG intensity contrast to the different symmetry properties between $\text{Cs}_3\text{Bi}_2\text{Br}_9$ samples with odd- and even-layer. SHG emissions show very bright signal in region 2, which may be due to the 3R-like stacking with the vertical stacking angle $\theta=0^\circ$ (Figure R2e). The total SHG intensity in stacking region can be expressed as:

$$I_S(\theta) = I_1 + I_2 + 2\sqrt{I_1 I_2} \cos 3\theta$$

where I_S , I_1 , and I_2 stand for the SHG intensity in the stacking region, the 6-layered $\text{Cs}_3\text{Bi}_2\text{Br}_9$ and the 7-layered $\text{Cs}_3\text{Bi}_2\text{Br}_9$, respectively. θ is the stacking angle ($\theta=0^\circ$). The SHG intensity in the stacking region is well fitted with this equation (Figure R7e). This new phenomenon is the first observation in the field of metal halide materials. These SHG findings on atomically thin halide perovskites would be beneficial for fundamental research and potential application in the field of electronics and optoelectronics.

Figure R7 (a) The optical images of the stacking 6-layered and the 7-layered $\text{Cs}_3\text{Bi}_2\text{Br}_9$. (b) The SHG intensity mapping of the region in a. (c) The AFM images of the region in a. (d) The height of the marked line in region a. (e) The SHG intensity of the three regions. (f) The diagram with three regions with stacking angle of $\theta=0^\circ$.

3. The authors claim that the colloidal particles are thermodynamically unstable and can release metal salt ions for the growth of micro-platelets. What is the content of the colloidal particles. It is also possible that the particles aggregate to form the micro-platelets.

A: According to the reviewer's concerns, we have added experiments and discussion about colloidal solution in revised manuscript (Page 4, Line 26-29; Page 5, Line 1-7). The diagram of the Stern double layer (typical colloids' structure) is shown in Figure R8a. The nucleus particle could be negatively charged, attracting positive ions to form the Stern layer, where ions are strongly bound to nucleus

particle. The diffusion layer contains a large amount of solvent and ions, and the ions are loosely attached the Stern layer. We carried out the zeta potential of $\text{Cs}_3\text{Bi}_2\text{Br}_9$ colloidal solution to understand the content and mechanism of colloidal solution during growth (Figure R8b). While temperature increases, the ions source of colloidal solution continually release for the growth of the atomically thin $\text{Cs}_3\text{Bi}_2\text{Br}_9$ on the substrate with the help of surfactants.

Figure R8 (a) The Stern double layer model of the colloidal solution. (b) The zeta potential of $\text{Cs}_3\text{Bi}_2\text{Br}_9$ colloidal solution with varied temperature from 20°C to 90°C.

As for the colloidal solution, we can estimate content and ion quantity in the colloidal particle of the solution. According to the specific surface area ($1.20 \times 10^4 \text{ m}^2 \text{ kg}^{-1}$ measured by laser particle size analyzer) and the particle size ($0.153 \text{ }\mu\text{m}$ measured by laser particle size analyzer) of the colloid solution, the content of the colloid particle can be calculated as $2.60 \times 10^{-7} \text{ mol/L}$. The charge density in the diffuse layer can be obtained from the nonlinear Poisson-Boltzmann equation (Eq. 3) (Electrochemical Methods: Fundamentals and Applications, second ed., Wiley, New York, 2001):

$$\sigma = -(8kT\varepsilon\varepsilon_0c_{bulk})^{\frac{1}{2}} \sinh\left(\frac{\zeta e\phi_0}{2kT}\right) \quad (3)$$

Where σ is the charge density, ϕ_0 is total potential drop across the solution side of the double layer ($\phi_0 >$ zeta potential, approximating ϕ_0 to zeta potential), ε is dielectric constant of relative permittivity, ε_0 is the permittivity of the vacuum, ζ is the valence of species, c_{bulk} is the concentration of solution (12.3 mmol/L by ICP measurement, Figure R9). The charge density σ can be calculated as $3.14 \times 10^{-4} \text{ C m}^{-2}$. The diffusion layer charge of single colloidal particle is about $2.31 \times 10^{-17} \text{ C}$ (The colloidal particle size is $0.153 \text{ }\mu\text{m}$ measured by laser particle size analyzer). During the experimental process, the whole solution keeps clear without any turbid phenomena and there are no

any aggregate particles on the substrate. Under the observation of electronic microscopy, the atomically thin $\text{Cs}_3\text{Bi}_2\text{Br}_9$ presents a single-crystalline nature without grain boundary, suggesting the nucleation growth process not particles aggregate on the substrate.

Figure R9 The concentration of supersaturation state as a function of the n-OA variation.

4. How does the substrate affect the synthesis?

A: According to the reviewer's suggestion, we have included more experiments to investigate the growth of atomically thin $\text{Cs}_3\text{Bi}_2\text{Br}_9$ on different substrates in revised manuscript (Page 5, Line 23-29; Page 6, Line 1-3). The different substrates have an influence on the size and thickness (Figure R10), which might be caused by lattice match. As it is well known, the lattice spacing values of in Al_2O_3 along the $\langle 10\bar{1}0 \rangle$ direction ($\frac{\sqrt{3}}{2} a_{\text{Al}_2\text{O}_3} = 0.412 \text{ nm}$), which can match well with the lattice spacing values in $\text{Cs}_3\text{Bi}_2\text{Br}_9$ ($a_{\text{Cs}_3\text{Bi}_2\text{Br}_9} = 0.822 \text{ nm}$) by building the supercells. The experimental results show the $\text{Cs}_3\text{Bi}_2\text{Br}_9$ on sapphire substrate of well lattice match tends to smaller thickness and larger aspect ratios (Figure R10c-c'). In general, the heterogeneous nuclei can bind more strongly to the substrate. Heterogeneous nucleation is promoted on substrate, while homogeneous nucleation is suppressed, resulting in atomically thin film being formed on the substrate surface layer-by-layer (*Chem. Soc. Rev.* **2019**, 48, 2011). In addition, we find that the wettability of the substrate also affects the growth of atomically thin $\text{Cs}_3\text{Bi}_2\text{Br}_9$ nanosheets. The atomically thin $\text{Cs}_3\text{Bi}_2\text{Br}_9$ grown on ozone-treated substrates exhibits larger lateral sizes and aspect ratios compared to those grown on untreated substrates (Figure R10a'-d').

Figure R10 The atomically thin $\text{Cs}_3\text{Bi}_2\text{Br}_9$ growing on untreated (a) mica, (b) SiO_2 , (c) Si and (d) sapphire substrates. The atomically thin $\text{Cs}_3\text{Bi}_2\text{Br}_9$ growing on ozone-treated (a') mica, (b') SiO_2 , (c') Si and (d') sapphire substrates.

Figure R11a-d presents the wettability of the untreated mica, SiO_2 , Si and sapphire substrates. Obviously, the ozone-treated substrates have smaller contact angles than these of untreated substrates (Figure R11a'-d'). Figure R11e illustrates an interfacial energy diagram for three phases of two solids and a liquid in contact. The terms γ_{cl} , γ_{sl} and γ_{cs} represent interfacial energies between the crystal phase and the liquid, the solid surface and the liquid, and the crystal phase and the solid surface, respectively. The nuclei and active centers have a high affinity with the solid surface. And the nucleation barrier (ΔG_{hetero}) is dramatically lowered due to effective reduction in the interface energy. The free energy for heterogeneous nucleation is corrected by introducing Φ , which is the ratio of free energies of homogeneous and heterogeneous nucleation (Eq. R1). Φ is a factor dependent on the contact angle θ (Eq. R2).

$$\Delta G_{hetero} = \Phi \Delta G_{hetero} \quad (\text{R1})$$

$$\Phi = \frac{(2 + \cos\theta)(1 - \cos\theta)^2}{4} \quad (\text{R2})$$

The relationship between Φ and θ indicates that heterogeneous nucleation readily occurs at a low concentration state (Figure R4f). In order to grow atomically thin $\text{Cs}_3\text{Bi}_2\text{Br}_9$ with large aspect ratio, it is important to form heterogeneous nuclei on the substrate rather than homogeneous nucleation (*Chem. Soc. Rev.* **2019**, 48, 2011). As a result, the hydrophilic substrate enhances heterogeneous nucleation to produce bigger lateral sizes of atomically thin $\text{Cs}_3\text{Bi}_2\text{Br}_9$.

Figure R11 The wettability of the untreated (a) mica, (b) SiO₂, (c) Si and (d) sapphire substrates. The wettability of the ozone-treated (a') mica, (b') SiO₂, (c') Si and (d') sapphire substrates. (e) Illustration of the contact angle θ for heterogeneous nucleation. (f) Ratio of free energies of homogeneous and heterogeneous nucleation Φ as a function of the contact angle θ .

5. How to know the surfactant (carboxyl group) is binding with the Br atom based on the XPS and IR spectrum (Figure S2 and S3), please give detailed explanation.

A: Thanks for your comment. The detailed explanation about X-ray photoelectron spectroscopy (XPS) and Fourier transform infrared spectroscopy (FTIR) spectrum has been added in revised manuscript (Page 5, Line 8-11). We used XPS and FTIR to investigate the interaction between n-OA and Cs₃Bi₂Br₉. Figure R12 shows the XPS of Cs 3d, Bi 4f and Br 3d spectra. There is no evident shift in the Cs 3d peak (R12a). Compared with Cs₃Bi₂Br₉, Cs₃Bi₂Br₉+n-OA sample shows that core-level peaks of Bi 4f (Figure R12b) and Br 3d (Figure R12c) are shifted towards low binding energy. This demonstrates that the C=O moiety donates its lone electron pair on the oxygen atoms to the empty 6p orbital of Bi³⁺ (*Nat. Commun.* **2021**, 12, 1246), not only decreasing the cationic charge but also leading to a change in the electrostatic interaction between the Bi³⁺ and the Br⁻ ions (*Nature* **2021**, 599, 594). In the FTIR spectra, we observed an infrared peak at 1720 cm⁻¹ arising from the C=O of -COOH group stretching vibration

for n-OA (*Nat. Energy* **2016**, 1, 16142). The C=O peak shifts to a lower wavenumber of 1710 cm^{-1} for $\text{Cs}_3\text{Bi}_2\text{Br}_9$ +n-OA sample, indicating a weakened C=O bond strength caused by the interaction (*Nat. Energy* **2016**, 1, 16142) (Figure R13). The -OH of -COOH group rocking vibration for n-OA is found that this shifted from 953 cm^{-1} to a lower wavenumber 933 cm^{-1} for $\text{Cs}_3\text{Bi}_2\text{Br}_9$ +n-OA sample, indicating that there is a slight interaction between n-OA and $\text{Cs}_3\text{Bi}_2\text{Br}_9$ on -COOH group. As for the organic acids, the binding of carboxyl group -COOH ligands usually involves hydrogen bonding between the protons located on the -COOH ligand's anchoring group -OH and the Br anions, with -COOH ligand's headgroup -C=O interacting with positive Bi sites, respectively (*ACS Nano* **2022**, 16, 1444).

Figure R12 XPS (a) Cs 3d, (b) Bi 4f and (c) Br 3d core level patterns of the $\text{Cs}_3\text{Bi}_2\text{Br}_9$ that synthesis with and without n-OA.

Figure R13 Infrared spectrum spectra from 2000 cm^{-1} to 500 cm^{-1} of n-OA, $\text{Cs}_3\text{Bi}_2\text{Br}_9$ with and without n-OA.

Reviewer #3:

The work "Colloid Driven Low Supersaturation Crystallization for Atomically Thin Bismuth Halide Perovskite Single Crystal Films" reports on the synthesis and characterization of Cs₃Bi₂Br₉ thin platelets. The authors demonstrated that, by tuning the amount of n-octanoic acid (n-OA) in the reaction mixture, they could tune the lateral size and the thickness of the obtained platelets. Higher amount of n-OA allows to obtain single-cell-thick crystals.

The authors demonstrated also that the material displays non-linear optical properties.

The manuscript has several flaws, in my opinion.

First, it is difficult to read - it might require a deep revision of the used English.

A: Thanks for your advice. The English of the whole manuscript has been polished by professional service provider Spring Nature as shown in Figure R14.

Figure R14 Editing certificate for our manuscript by Spring Nature.

Second, it is hard to identify the specific novelty of the work (maybe because of the first issue on language): reports on the colloidal synthesis of Bi-based "perovskites" and on their nonlinear optical properties can be found already in literature. The authors should highlight better how their approach differs from those of other research groups.

A: We apologize for unclear English expression of the work in the original version. The revised version

of the whole manuscript has been polished with the assistance of English professional service provider Spring Nature.

According to the reviewer's suggestion, the revised manuscript has clearly presented the growth process of the atomically thin $\text{Cs}_3\text{Bi}_2\text{Br}_9$ (Page 4, Line 23-29; Page 5, Line 1-12) and new findings on nonlinear optical properties (Page 9, Line 3-29; Page 10, Line 1-3) as well as the comparison with other works (Page 5, Line 20-23). Herein, our work is not a colloidal synthesis of Bi-based perovskites, but new colloidal strategy for a heterogeneous nucleation substrate-dependent growth process of atomically thin halide perovskite with high aspect ratio. This colloidal solution is used to build up a low supersaturation solution, which means low concentration of ions in bulk solution and more ions in the colloid (Figure R15). Low concentration of ions aims to be no nucleation in the bulk solution but possible nucleation on the substrate, and supersaturation could continually provide ions' source from colloids for the atomically thin $\text{Cs}_3\text{Bi}_2\text{Br}_9$ growth on the substrate. During the experiment process, we observe an obvious Tyndall effect (Figure R15a). Based on the typical colloids' structure described by the Stern double layer model (Electrochemistry Principles, Methods and Applications, *Oxford Univ. Press*, Oxford, **1993**; *J. Colloid Interf. Sci.* **2007**, 305, 159), Figure R15b shows the colloidal diagram of the Stern double layer. The nucleus particle could be negatively charged, attracting positive ions to form the Stern layer, where ions are strongly bound to nucleus particle. The diffusion layer contains a large amount of solvent and ions, and the ions are loosely attached the Stern layer. The diffusion layer can continuously exchange ions with the bulk solution. According to the zeta potential, the colloid system is stable (Figure R5c). As a result, the very low concentration of ions remains in bulk solution, many ions are included in the colloidal structure. As-prepared low supersaturation solution is aimed to reduce the nucleation in bulk solution, and enhance the heterogeneous nucleation on the substrate. While temperature increases, the ions source of colloidal solution continually release for the growth of the atomically thin $\text{Cs}_3\text{Bi}_2\text{Br}_9$ on the substrate with the help of surfactants. In summary, the atomically thin $\text{Cs}_3\text{Bi}_2\text{Br}_9$ growth is substrate-dependent heterogeneous nucleation process. The colloid solution provides a stable precursor source and create a low-concentration solute system. It drives a low-concentration solute to initialize heterogeneous nucleation on the substrate not in the bulk solution. As a result, the atomically thin $\text{Cs}_3\text{Bi}_2\text{Br}_9$ could be achieved with larger lateral sizes and aspect ratios. In comparison with the previous reports of atomically thin halide perovskite in Figure R16, our $\text{Cs}_3\text{Bi}_2\text{Br}_9$ has the largest aspect ratio (size/thickness) over 10^4 , which is beneficial for the

application of atomically thin halide perovskite materials in optoelectronic devices and electronic device integration.

Figure R15 (a) The Tyndall effect of original solution and colloidal solution with adding n-OA. (b) The Stern double layer model of the colloidal solution. (c) The zeta potential of $\text{Cs}_3\text{Bi}_2\text{Br}_9$ colloidal solution in H_2O at 25°C .

Figure R16 The representative reports of atomically thin halide perovskite single crystals (References: 1. *Science* **2015**, 349, 6255; 2. *Nature* **2020**, 580, 614; 3. *Nat. Nanotechnol.* **2021**, 16, 584; 4. *Nat. Commun.* **2018**, 9, 5354; 5. *ACS Energy Lett.* **2020**, 5, 1900; 6. *Chem. Mater.* **2020**, 32, 721; 7. *ACS Nano* **2019**, 13, 14294; 8. *Angew. Chem.* **2017**, 56, 14187; 9. *Adv. Mater.* **2020**, 32, 200411; 10. *Nano Res.* **2021**, 14, 4079; 11. *ACS Nano* **2020**, 14, 11294).

The clear diagram of growth process can be described in Figure R17. Firstly, the original precursor solution is uniformly distributed (Figure R17a). The n-OA is added in the original solution to form a colloidal solution with Tyndall effect. According to Fajans rule, the most of precursor ion is absorbed in the colloids and less ions remain in the bulk solution (Figure R17b). The low supersaturation

solution is aimed to reduce the nucleation in the solution, and enhance the heterogeneous nucleation on the substrate. With the increase of the temperature, the heterogeneous nucleation occurs on the substrate (Figure R17c). Simultaneously, the adsorption strength of the ions of colloidal solution is weakened. The ions absorbed by the colloids have higher kinetic energy to move to the surrounding heterogeneous nuclei on the substrate. That is, colloid solution provides the precursor ion source in the low-concentration solute system to grow on the substrate. Moreover, the surfactant is directionally adsorbed on the $\text{Cs}_3\text{Bi}_2\text{Br}_9$ specific crystal facets to regulate the surface energy and promote anisotropic growth (Figure R17d). Finally, the atomically thin $\text{Cs}_3\text{Bi}_2\text{Br}_9$ are grown on the substrate (Figure R17e).

Figure R17 Illustration of the colloid driven low supersaturation crystalline for atomically thin single crystal $\text{Cs}_3\text{Bi}_2\text{Br}_9$ growth. (a) The original solution of $\text{Cs}_3\text{Bi}_2\text{Br}_9$. (b) The n-OA is added in the original solution to form a colloidal solution with Tyndall effect. (c) The heterogeneous nucleation on the substrate. (d) The growth process on the substrate. (e) Atomically thin $\text{Cs}_3\text{Bi}_2\text{Br}_9$ with ultra-high aspect ratio is obtain.

Except for the normal second harmonic generation of metal halide materials, we found some new SHG phenomena and interesting results on nonlinear optical properties of atomically thin $\text{Cs}_3\text{Bi}_2\text{Br}_9$. Atomically thin $\text{Cs}_3\text{Bi}_2\text{Br}_9$ shows layer-dependent SHG intensity (Figure R18) and stacking-induced SHG intensity (Figure R19). The vertical stacking phenomenon is the first observation in the field of metal halide materials. The specific experiment and analysis are addressed as followings.

We have added new second harmonic generation (SHG) measurements to explore mutations of photophysical properties in revised manuscript (Page 9, Line 3-29; Page 10, Line 1-3). Atomically thin layers may exhibit mutations of photophysical properties, particularly as the number of layers reduces to a single layer. SHG measurements were carried out by the atomically thin $\text{Cs}_3\text{Bi}_2\text{Br}_9$ to explore the photophysical properties. Due to the three-fold rotational symmetry in the few-layer $\text{Cs}_3\text{Bi}_2\text{Br}_9$ crystal

structure, the co-polarized SHG intensity displays a six-fold rotational symmetry as a function of $\text{Cs}_3\text{Bi}_2\text{Br}_9$'s azimuthal angle θ (Figure R6a). We systematically measured SHG intensity for $\text{Cs}_3\text{Bi}_2\text{Br}_9$ crystals with 1-10 layers. SHG intensity exhibits different symmetry properties between the odd- and even-layer. Strong SHG signal can be detected for all odd-layers, while even-layers show weak SHG signal (Figure R18b). It is because that even-layers of $\text{Cs}_3\text{Bi}_2\text{Br}_9$ belong to the centrosymmetric D_{3d} space group, whereas odd-layer number belongs to the non-centrosymmetric D_{3h} space group. In addition, the SHG signal is reduced with the increase of the odd-layer number, which may be attributed to interlayer coupling. The variation of SHG intensity with odd- and even-layers shows the potential application in nonlinear optics for atomically thin $\text{Cs}_3\text{Bi}_2\text{Br}_9$.

Figure R18 (a) Polarization-resolved SHG spectra of the a three-layered $\text{Cs}_3\text{Bi}_2\text{Br}_9$, revealing its underlying threefold symmetry. (b) Layer-dependent SHG verified the systematically stronger signals from odd-layer $\text{Cs}_3\text{Bi}_2\text{Br}_9$ due to inversion symmetry breaking than from even-layer $\text{Cs}_3\text{Bi}_2\text{Br}_9$.

As it is well known, artificial structures created by stacked two-dimensional crystals have been the focus of intense research activity (*Nature* **2013**, 499, 419). As for twisted or stacked graphene layers (*Nature* **2013**, 497, 598) and transition metal dichalcogenides layers (*Science* **2022**, 376, 973), these structures can show unusual behaviors and new physical properties. Among the various layered compounds exhibit interesting properties governed by their structural symmetry and interlayer coupling, which are highly susceptible to stacking (*Nat. Nanotechnol.* **2014**, 9, 825). Herein, we found new and interesting results. Figure R19a-d shows very weak SHG intensity of the region 1 (6-layered $\text{Cs}_3\text{Bi}_2\text{Br}_9$), while the region 3 (7-layered $\text{Cs}_3\text{Bi}_2\text{Br}_9$) has higher SHG intensity. This may be caused by the SHG intensity contrast to the different symmetry properties between $\text{Cs}_3\text{Bi}_2\text{Br}_9$ samples with odd-

and even-layer. SHG emissions show very bright signal in region 2, which may be due to the 3R-like stacking with the vertical stacking angle $\theta=0^\circ$ (Figure R19e). The total SHG intensity in stacking region can be expressed as

$$I_S(\theta) = I_1 + I_2 + 2\sqrt{I_1 I_2} \cos 3\theta$$

where I_S , I_1 , and I_2 stand for the SHG intensity in the stacking region, the 6-layered $\text{Cs}_3\text{Bi}_2\text{Br}_9$ and the 7-layered $\text{Cs}_3\text{Bi}_2\text{Br}_9$, respectively. θ is the stacking angle ($\theta=0^\circ$). The SHG intensity in the stacking region is well fitted with this equation (Figure R19e). This new phenomenon is the first observation in the field of metal halide materials. These SHG findings on atomically thin halide perovskites would be beneficial for fundamental research and potential application in the field of electronics and optoelectronics.

Figure R19 (a) The optical images of the stacking 6-layered and the 7-layered $\text{Cs}_3\text{Bi}_2\text{Br}_9$. (b) The SHG intensity mapping of the region in a. (c) The AFM images of the region in a. (d) The height of the marked line in region a. (e) The SHG intensity of the three regions. (f) The diagram with three regions with stacking angle of $\theta=0^\circ$.

Apart from these two main issues, there also other problems:

A: Thank you for your constructive comments. We have added more experiments to improve the manuscript according to the reviewer's request.

1. The authors used the word "film" to describe their platelets.

A: According to the reviewer's comment, we have removed the word "film" in the revised manuscript. Herein, the atomically thin $\text{Cs}_3\text{Bi}_2\text{Br}_9$ is substrate-dependent heterogeneous nucleation process. The colloid solution provides a stable precursor source and creates a low-concentration solute system. It drives a low-concentration solute to initial heterogeneous nucleation on the substrate. If there are enough ions and time, the atomically thin $\text{Cs}_3\text{Bi}_2\text{Br}_9$ would be possibly obtained to cover the whole substrate. In comparison with the representative reports of atomically thin halide perovskite single crystals, our $\text{Cs}_3\text{Bi}_2\text{Br}_9$ has large aspect ratio (size/thickness) over 10^4 , which the largest one in the previous reports of Figure R16. Indeed, as-grown sample is not traditional film coating on the substrate, and is more similar to MoS_2 by chemical vapor deposition, which is usually called nanosheet or nanoplate. Therefore, we have changed the description of "film" in the revised manuscript.

2. The majority of the paper is focused on discussing concepts as supersaturation in colloidal synthesis, that might most likely be skipped.

A: According to the reviewer's suggestion, we have revised by emphasizing description about supersaturation (Page 5, Line 1-7). The colloidal system is used to build up a low supersaturation solution, which means low concentration of ions in bulk solution and more ions in the colloid. Low concentration of ion aims to be no nucleation in the bulk solution but possible nucleation on the substrate, and supersaturation could continually provide ions' source from colloids for the atomically thin $\text{Cs}_3\text{Bi}_2\text{Br}_9$ growth. The low supersaturation solution is aimed to enhance the heterogeneous nucleation on the substrate. Supersaturation can keep the film continuously growing and prevent atomically thin sample from dissolving. The low supersaturation solution can provide the continuous ions from colloidal solution to grow the atomically thin $\text{Cs}_3\text{Bi}_2\text{Br}_9$. The detailed process can be presented as the followings. The clear diagram of growth process can be described in Figure R17. Firstly, the original precursor solution is uniformly distributed (Figure R17a). The n-OA is added in the original solution to form a colloidal solution with Tyndall effect. According to Fajans rule, the most of precursor ion is absorbed in the colloids and less ions remain in the bulk solution (Figure R17b). The low supersaturation solution is aimed to reduce the nucleation in the solution, and enhance the heterogeneous nucleation on the substrate. With the increase of the temperature, the heterogeneous nucleation occurs on the substrate (Figure R17c). Simultaneously, the adsorption strength of the ions

by the charged particle is weakened. The ions absorbed by the colloids have higher kinetic energy to move to the surrounding heterogeneous nuclei on the substrate. That is, colloid solution provides the precursor source in the low-concentration solute system to grow on the substrate. Moreover, the surfactant is directionally adsorbed on the Cs₃Bi₂Br₉ specific crystal facets to regulate the surface energy and promote anisotropic growth (Figure R17d). Finally, the atomically thin Cs₃Bi₂Br₉ single crystals are grown on the substrate (Figure R17e).

3. The authors claim that carboxyl functional groups bind to Br atoms, based on XPS. It's hard to imagine how two negatively-charged ions are kept together.

A: Thanks for your comment. Herein, as for the organic acids, the binding of carboxyl group -COOH ligands usually involves hydrogen bonding between the protons located on the -COOH ligand's anchoring group -OH and the Br anions, with -COOH ligand's headgroup -C=O interacting with positive Bi sites, respectively. This phenomenon has been also reported in the previous reference (*ACS Nano* **2022**, 16, 1444). The detailed explanation about XPS and FTIR spectrum has been added in revised manuscript (Page 5, Line 8-11). We used XPS and FTIR to investigate the interaction between n-OA and Cs₃Bi₂Br₉. Figure R20 shows the XPS of Cs 3d, Bi 4f and Br 3d spectra. There is no evident shift in the Cs 3d peak (R20a). Compared with Cs₃Bi₂Br₉, Cs₃Bi₂Br₉+n-OA sample show that core-level peaks of Bi 4f (Figure R20b) and Br 3d (Figure R20c) are shifted towards low binding energy. This demonstrates that the C=O moiety donates its lone electron pair on the oxygen atoms to the empty 6p orbital of Bi³⁺ (*Nat. Commun.* **2021**, 12, 1246), not only decreasing the cationic charge but also leading to a change in the electrostatic interaction between the Bi³⁺ and the Br⁻ ions (*Nature* **2021**, 599, 594). In the FTIR spectra, we observed an infrared peak at 1720 cm⁻¹ arising from the C=O of -COOH group stretching vibration for n-OA (*Nat. Energy* **2016**, 1, 16142). The C=O peak shifts to a lower wavenumber of 1710 cm⁻¹ for Cs₃Bi₂Br₉+n-OA sample, indicating a weakened C=O bond strength caused by the interaction (*Nat. Energy* **2016**, 1, 16142) (Figure R21). The -OH of -COOH group rocking vibration for n-OA found that this shifted from 953 cm⁻¹ to a lower wavenumber 933 cm⁻¹ for Cs₃Bi₂Br₉+n-OA sample, indicating that there is little interaction between n-OA and Cs₃Bi₂Br₉ on -COOH group.

Figure R20 XPS (a) Cs 3d, (b) Bi 4f and (c) Br 3d core level patterns of the $\text{Cs}_3\text{Bi}_2\text{Br}_9$ that synthesis with and without n-OA.

Figure R21 Infrared spectrum spectra from 2000 cm^{-1} to 500 cm^{-1} of n-OA and $\text{Cs}_3\text{Bi}_2\text{Br}_9$ with and without n-OA.

4. Is the thickness scale for the AFM images of Figure 1 the same for all panels?

A: Yes, AFM images of Figure 1 has the same thickness scale for all panels, and this description has been added in the revised manuscript (Page 20, Line 4-5).

5. The histograms in Figure 2 are not discussed in the text nor in the figure caption. Lateral size and thickness statistics suggest moreover that the used method produces objects non really homogeneous in size.

A: According to the reviewer's comment, we have carried out more experiments and updated the Figure 2 for the optical microscope photograph by further optimization (Page 6, Line 13-24). The regulation of the temperature can only make the thickness of atomically thin single crystal $\text{Cs}_3\text{Bi}_2\text{Br}_9$ within about 10 nm. The further regulation of atomically thin $\text{Cs}_3\text{Bi}_2\text{Br}_9$ can be realized by subtle control of the surfactant concentration ($R_{\text{n-OA}} = \text{n-OA}/(\text{n-OA} + \text{DMSO})$) from 35% to 50% (Figure R22a-

h). In addition, the growth substrate has been optimized to sapphire substrate with lattice match and small contact angle θ , which is helpful to heterogeneous nucleation on substrate. The lateral size and thickness statistics are shown in Figure R22i. As the amount of surfactant increases, the thickness of the atomically thin single crystal $\text{Cs}_3\text{Bi}_2\text{Br}_9$ decreases significantly, and the lateral size of $\text{Cs}_3\text{Bi}_2\text{Br}_9$ also slightly decreases. It is helpful to achieve a large aspect ratio of atomically thin $\text{Cs}_3\text{Bi}_2\text{Br}_9$. At a concentration of 40%, the thickness of $\text{Cs}_3\text{Bi}_2\text{Br}_9$ can be achieved below 5 nm, and the lateral size can reach over 50 μm , which presents a high aspect ratio over 10^4 . The large lateral size, combined with the atomically thin thickness, is beneficial for further application for optoelectronic devices and device integration. While the amount of surfactant over 45%, it is hard to obtain complete hexagonal atomically thin $\text{Cs}_3\text{Bi}_2\text{Br}_9$ due to the low precursor concentration.

Figure R22 Optical microscopy images of the atomically thin $\text{Cs}_3\text{Bi}_2\text{Br}_9$ that produced from ranged n-OA percentage: (a) 35%, (b) 37%, (c) 39%, (d) 40%, (e) 41%, (f) 43%, (g) 45%, and (h) 50%. (i) The statistics of $\text{Cs}_3\text{Bi}_2\text{Br}_9$ thicknesses and lateral sizes as a function of ranged n-OA percentage.

REVIEWERS' COMMENTS

Reviewer #1 (Remarks to the Author):

The revised version of the manuscript is suitable for publication.

Reviewer #2 (Remarks to the Author):

The manuscript is modified according to the comments. Now the material is named as atomically thin bismuth halide perovskite, which is reasonable. Compared to the previous reports of atomically thin halide perovskite, the Cs₃Bi₂Br₉ possesses the largest aspect ratio (size/thickness) over 104, which is beneficial for the application of atomically thin halide perovskite materials in optoelectronic devices and electronic device integration. Therefore, I recommend publishing it.

Reviewer #3 (Remarks to the Author):

Even if the authors worked at the manuscript revision, the paper is still extremely qualitative and does not provide sufficient evidences of novelty to be accepted in Nature Communications. I therefore recommend its rejection.

RESPONSE TO REVIEWERS' COMMENTS

Reviewer #1 (Remarks to the Author):

The revised version of the manuscript is suitable for publication.

A: Many thanks for the Reviewer #1's positive comments.

Reviewer #2 (Remarks to the Author):

The manuscript is modified according to the comments. Now the material is named as atomically thin bismuth halide perovskite, which is reasonable. Compared to the previous reports of atomically thin halide perovskite, the $\text{Cs}_3\text{Bi}_2\text{Br}_9$ possesses the largest aspect ratio (size/thickness) over 10^4 , which is beneficial for the application of atomically thin halide perovskite materials in optoelectronic devices and electronic device integration. Therefore, I recommend publishing it.

A: Many thanks for the Reviewer #2's compliments.

Reviewer #3 (Remarks to the Author):

Even if the authors worked at the manuscript revision, the paper is still extremely qualitative and does not provide sufficient evidences of novelty to be accepted in Nature Communications. I therefore recommend its rejection.

A: Many thanks for the Reviewer #3's comments. As for the experimental supplementary, we have provided sufficient evidences to demonstrate the innovation of this work. For example, many data are collected to for the first time report a colloid-driven low supersaturation crystallization strategy for growth of atomically thin $\text{Cs}_3\text{Bi}_2\text{Br}_9$. Moreover, direct experiments demonstrate the atomically thin $\text{Cs}_3\text{Bi}_2\text{Br}_9$ has layer-dependent nonlinear optical performance and stacking-induced second harmonic generation. Specially, the vertical stacking phenomenon presents the interesting observation of enhanced signal for second harmonic generation emission in the field of metal halide materials. The Reviewer #3 did not raise any specific suggestions and objections about our previous revisions this time, so he/she should be satisfied with our revised manuscript. In summary, this work provides a new concept for growing atomically thin halide perovskite with non-van der Waal structures and demonstrates the application for atomically thin single crystals' growth with strong electronic coupling between adjacent layers.

Brief Summary

It is challenging to grow atomically thin non-van der Waals perovskite due to the strong electronic coupling between adjacent layers. Here, Li et al. established a colloid-driven low supersaturation crystallization strategy to grow atomically thin bismuth halide perovskite.